# Primary Prophylaxis Lapelga® in Early Breast Cancer: A Real-World Experience

**Fahad Khan [1], Morgan Black [2] , Alaina Charlton [2] and Jawaid Younus [2],***

[1] Department of Medicine, Schulich School of Medicine and Dentistry, Western University, London, ON N6A 5C1, Canada

[2] Department of Oncology, London Regional Cancer Program, London Health Sciences Centre, London, ON N6A 5W9, Canada

* Correspondence: jawaid.younus@lhsc.on.ca

**Abstract:** Background: Lapelga® was approved by Health Canada as a pegfilgrastim biosimilar in 2019 and remains the most commonly used biosimilar in Ontario and is fully reimbursed under the Ontario Drug Benefit program in this category. We explored the efficacy and tolerability of Lapelga® in a retrospective analysis of patients with early breast cancer who underwent adjuvant chemotherapy supported with Lapelga® as a primary prophylaxis. Methods: Adult patients with early breast cancer treated with adjuvant chemotherapy at the London Regional Cancer Program in London, ON, Canada between May 2019 and June 2022 were included. All of these patients were supported with Lapelga® as the primary prophylaxis. Patients' age, tumour, and nodal status, their type of chemotherapy, co-morbid conditions, and incidence of febrile neutropenia (FN) and its related details as well as any reported side effects to Lapelga® were collected. Results: A total of 201 patients were included in this review with majority (78%) of patients under 65 years of age. One third of patients were treated with the adriamycin and cyclophosphamide (AC)-Paclitaxel dose dense chemotherapy and a quarter of patients with either a docetaxel and cyclophosphamide (TC) combination or an AC-dose dense with Paclitaxel weekly, and 10% or less patients had FEC-D (5-fluorouracil, epirubicin, and cyclophosphamide) and AC chemotherapy. FN incidence was only 3.48% in this review (7/201 patients). Patients with FN were admitted to hospital and recovered completely with no mortality reported. No cases of a switch to a different granulocyte colony growth factor were seen. The most frequent side effects from Lapelga® included musculoskeletal pain, fever, and headache. However, the majority of patients (88.6%; 178/201) did not have any reported side effects specifically assigned to Lapelga®. Conclusions: In this single centre retrospective study, early breast cancer patients (n = 201) treated with adjuvant chemotherapy supported with primary prophylaxis with Lapelga® had a low incidence of FN (3.48%). This supports Lapelga® being an effective strategy as the primary prophylaxis when used with common chemotherapy regimens in the real-world setting.

**Keywords:** Lapelga®; pegfilgrastim; breast cancer; febrile neutropenia

## 1. Introduction

Adjuvant chemotherapy reduces the risk of recurrence and improves overall survival for the patients with early breast cancer [1]. Febrile neutropenia (FN) remains among the common toxicities related to the use of adjuvant chemotherapy with the potential for hospitalization and an increased risk of mortality [2,3]. The use of granulocyte colony-stimulating growth factors (G-CSF) has significantly reduced the incidence and impact of FN in these patients, and it is recommended by all guidelines to be used preferably as the primary prophylaxis [4,5]. The pegylated formulation of G-CSF offers a convenient, single, fixed-dose and appears to have an equivalent efficacy and side-effect profile as non-pegylated formulations [6]. Biosimilars have been introduced slowly into the clinical

management plans and refer to a medicinal product similar to a reference biological medicine that has already been approved. Although the concept of biosimilars are close to the generic products, their production and approval differ markedly. Biosimilar approval will require a number of steps to ensure structural and functional similarity to the reference molecule. However, the clinical trials to test their efficacy head-to-head against the reference biologic may not be an essential step in the approval process. In such situations, the real-world experience and data may provide the missing pieces of information and an assurance of the efficacy of the biosimilar product.

In Ontario, Canada, the provincial reimbursement was granted for the first time to the pegylated G-CSF biosimilar (Lapelga®) in August 2019. We review here the early breast cancer patients treated with adjuvant chemotherapy at London Regional Cancer Program (London, Ontario, Canada), with Lapelga® used as the primary prophylaxis to gain real-world experience related to its efficacy and side effects.

## 2. Materials and Methods

All adult patients with early breast cancer treated with adjuvant chemotherapy and with Lapelga® as the primary prophylaxis at the London Regional Cancer Program between May 2019 and June 2022 were included in this retrospective review. This study was a part of the Breast Cancer Database Project and had an approval of retrospective case entries by the Ethics Committee at Western University, London, ON. Electronic medical records of these patients were reviewed and data parameters were extracted, which included demographic information, stage, co-morbid conditions, type of chemotherapy regimen, incidence of FN neutropenia, and recorded side effects related to Lapelga®. Our institution defines FN as a temperature of 38 °C with a neutropenia level of $0.5 \times 10^9$/L or below $1.0 \times 10^9$/L and expected to worsen.

## 3. Results

Our review included 201 adult patients with early breast cancer. Only 44 patients were over 65 years of age and the median age was 57 years (range: 28–79 years). These patients appeared to have aggressive pathological features, with the majority of patients having larger tumour size and/or lymph node involvement (Table 1). The type of chemotherapy used spanned across different combinations. AC-Paclitaxel was the most frequently used choice of chemotherapy, either in the dose dense or as a weekly Paclitaxel format, followed by TC (Docetaxel and Cyclophosphamide once every 3 weeks) chemotherapy. AC (Doxorubicin + Cyclophosphamide once every 3 weeks), AC-Docetaxel (Doxorubicin + Cyclophosphamide followed by Docetaxel once every 3 weeks), and FEC-D (Fluorouracil, Epirubicin, and Cyclophosphamide once every 3 weeks followed by Docetaxel once every 3 weeks) were used less frequently in this adjuvant setting (Table 2). The most frequently noted co-morbid conditions included hypertension, hypothyroidism, diabetes, and dyslipidemia. There were a total 92 patients (45.7%) with these noted co-morbid conditions (Table 3). At least one of these conditions was present in 40 patients (40/115; 34.8%) under 65 years and in 22 patients (22/41; 53.7%) who were aged 65 years or older.

Overall, FN was noted in only seven patients (3.48%). Their age, pathological features, and other details are summarized in Table 4. Three of these patients were treated with TC chemotherapy with FN developing in the very first cycle, and one with the AC-Docetaxel combination where FN was noted in the third cycle. The remaining two patients who experienced FN received AC dose dense paclitaxel weekly and developed FN in the first and third cycles, respectively. These patients were admitted to the hospital and had uneventful recovery. Only one patient discontinued use of Lapelga® with the other six patients having continued with Lapelga® in the remaining cycles of chemotherapy with dose reduction and without any further episode of FN. There were very few side effects reported related to the use of Lapelga®. Those reported included fever (without neutropenia), headache, pain at the injection site, and muscle aches.

**Table 1.** Pathological features of early breast cancer patient tumors included in this study.

| Tumour Size (T) | Number of Patients | Lymph Node (N) | Number of Patients |
|---|---|---|---|
| Tx | 1 | Nx | 4 |
| T1 | 59 | N0 | 78 |
| T2 | 95 | N1 | 100 |
| T3 | 34 | N2 | 13 |
| T4 | 12 | N3 | 6 |

Tx: Primary tumour cannot be assessed; T1: Tumour ≤ 2 cm in greatest dimension; T2: Tumour > 2 cm but ≤5 cm in greatest dimension; T3: Tumour > 5 cm in greatest dimension; T4: Tumor of any size with direct extension to the chest wall and/or to the skin (ulceration or skin nodules).; Nx: Regional lymph nodes cannot be assessed; N0: No regional lymph node metastases; N1: Metastases to movable ipsilateral level I, II axillary lymph node(s); N2: Metastases in ipsilateral level I, II axillary lymph nodes that are clinically fixed or matted; or in clinically detected (including imaging) ipsilateral internal mammary nodes in the absence of clinically evident axillary lymph node metastases; N3: Metastases in ipsilateral infraclavicular (level III axillary) lymph node(s) with or without level I, II axillary lymph node involvement; or in clinically detected (including imaging) ipsilateral internal mammary lymph node(s) with clinically evident level I, II axillary lymph node metastases; or metastases in ipsilateral supraclavicular lymph node(s) with or without axillary or internal mammary lymph node involvement.

**Table 2.** The number of patients who received each chemotherapy.

| Chemotherapy Regimen | Number of Patients |
|---|---|
| AC | 3 |
| AC-Doc | 7 |
| AC-Paclitaxel DD | 71 |
| AC-DD/weekly Paclitaxel | 52 |
| FEC-D | 16 |
| TC | 52 |

AC: Doxorubicin + Cyclophosphamide once every 3 weeks; AC-Doc: Doxorubicin + Cyclophosphamide followed by Docetaxel once every 3 weeks; AC-Paclitaxel DD: Doxorubicin + Cyclophosphamide followed by Paclitaxel all administered dose dense (every two weeks); AC-DD: Doxorubicin + Cyclophosphamide administered dose dense (every 2 weeks) followed by weekly Paclitaxel for 12 doses; FEC-D: Fluorouracil, Epirubicin, and Cyclophosphamide administered once every 3 weeks for 3 doses followed by Docetaxel once every 3 weeks for 3 doses; TC: Docetaxel + Cyclophosphamide once every 3 weeks.

**Table 3.** The number of patients with the most frequently observed co-morbidities.

| Most Frequent Co-Morbid Conditions | Number of Patients |
|---|---|
| Hypertension | 37 |
| Hypothyroidism | 27 |
| Diabetes | 16 |
| Dyslipidemia | 12 |

**Table 4.** Demographic, diagnostic, and treatment information for the three patients who developed febrile neutropenia while receiving chemotherapy.

| Age (Years) | Tumour (T) | Lymph Node (N) | Co-Morbidities | Chemotherapy | Cycle Number | Treatment | Mortality |
|---|---|---|---|---|---|---|---|
| 50 | T1 | N1 | Celiac disease, Fibromyalgia, Cluster headaches | TC | 1 | In-patient | No |
| 53 | T3 | N0 | Anxiety, Thyroid cancer | TC | 1 | In-patient | No |
| 59 | T1 | N0 | None | AC-Doc | 3 | In-patient | No |

**Table 4.** *Cont.*

| Age (Years) | Tumour (T) | Lymph Node (N) | Co-Morbidities | Chemotherapy | Cycle Number | Treatment | Mortality |
|---|---|---|---|---|---|---|---|
| 61 | T4c | N1 | Obstructive sleep apnea, Obesity | FEC-D | 5 | In-patient | No |
| 54 | T1c | N0 | Hypertension | TC | 1 | In-patient | No |
| 62 | T2 | N0 | Hypertension, Reflux disease | AC-DD/weekly Paclitaxel | 1 | In-patient | No |
| 53 | T3 | N1 | Hypertension | AC-DD/weekly Paclitaxel | 3 | In-patient | No |

T1: Tumour $\leq$ 2 cm in greatest dimension; T1c: Tumor > 1 cm but $\leq$ 2 cm in greatest dimension; T2: Tumour > 2 cm but $\leq$ 5 cm in greatest dimension; T3: Tumour > 5 cm in greatest dimension; T4c: Extension to the chest wall, not including only pectoralis muscle adherence/invasion AND Ulceration and/or ipsilateral satellite nodules and/or edema (including peau d'orange) of the skin, which do not meet the criteria for inflammatory carcinoma; N0: No regional lymph node metastases; N1: Metastases to movable ipsilateral level I, II axillary lymph node(s); TC: Docetaxel + Cyclophosphamide once every 3 weeks; AC-Doc: Doxorubicin +Cyclophosphamide followed by Docetaxel once every 3 weeks; FEC-D: Fluorouracil, Epirubicin, and Cyclophosphamide administered once every 3 weeks for 3 doses followed by Docetaxel once every 3 weeks for 3 doses; AC-DD: Doxorubicin + Cyclophosphamide administered dose dense (every 2 weeks) followed by weekly Paclitaxel for 12 doses.

## 4. Discussion

In this retrospective study, we report the real-world experience with the biosimilar pegfilgrastim (Lapelga®) when used as the primary prophylaxis to adjuvant chemotherapy for early breast cancer. In our 201 patient series, the incidence of FN was brought down to 3.48% with Lapelga®. This is in line with the reported efficacy of the parent pegfilgrastim molecule (Neulasta®) [7]. The side effects reported with Lapelga® were similar to Neulasta®.

Neulasta®, the originator pegfilgrastim manufactured by Amgen, was approved by Health Canada based on three phase III double-blinded randomized controlled trials of breast cancer patients [7]. In two head-to-head trials, patients with breast cancer received doxorubicin and docetaxel, followed by a G-CSF 24 hours later (either one of: Neulasta®, pegfilgrastim 6 mg once, or 100 mcg/kg once, or filgrastim 5 mcg/kg/day with a mean duration of 11 days) [8]. The results confirmed Neulasta®'s non-inferiority to filgrastim [8]. In terms of FN, 13% of patients that received Neulasta® experienced FN, compared to 20% with filgrastim ($-7$% difference; 95% CI of $-19$–+5%) [8]. Another study including both metastatic and non-metastatic breast cancer patients further evaluated its efficacy for primary prophylaxis of FN [9]. Patients received a cycle of docetaxel 100 mg/m$^2$ q3w for four cycles then 24 h afterwards received subsequent Neulasta® (pegfilgrastim 6 mg once or 100 mcg/kg once) or a placebo and allowed crossover for patients that experienced FN during the study [9]. The incidence of FN for patients that received Neulasta® was only 1% compared to 17% in the placebo arm ($p \leq 0.001$) [9]. This result was also associated with a significantly lower incidence of hospitalization and IV antibiotics in the Neulasta® group (both with $p \leq 0.001$) [9]. Based on the above evidence, Neulasta® was approved by Health Canada in 2004 [10]. However, the Ontario Drug Benefit (ODB), which pays for medication for seniors ($\geq$65 years old), children ($\leq$24 years old), long-term care residents, patients on disability, and patients receiving Ontario Works or home care [11], does not provide funding for Neulasta® [12].

Two meta-analyses performed in 2011 and 2015 compared patients receiving treatment for both solid tumours and lymphoma found FN to be significantly reduced when patients received G-CSFs (filgrastim, pegfilgrastim, lenograstim, and lipegfilgrastim) versus the placebo [13,14]. They further determined that pegfilgrastim significantly reduced FN risk when used as the primary prophylaxis in comparison to filgrastim [13,14].

Lapelga®, the first pegfilgrastim biosimilar approved in Canada, was required to undergo a comparative bioavailability study in order to obtain Health Canada approval [15]. In this study, Neulasta® and Lapelga® were compared in terms of bioavailability and the

ratios of geometric means, which for the test (Lapelga®)/reference (Neulasta®) were all found to be within the acceptable range, as defined for all biosimilars, with the relevant pharmacokinetic parameters within 80–125% of the originator [16]. The pharmacodynamic parameter of the absolute neutrophil count (ANC) was also within the defined parameter [16]. Upon achieving these parameters, Health Canada approved Apotex Inc.'s Lapelga® in 2019 [15]. Lapelga® was granted approval by ODB for reimbursement on 30 August 2019 [12], which allowed for a larger number of patients to access it for primary prophylactic treatment of FN.

Based on the evidence that pegfilgrastim reduces FN effectively [13,14], and given the recent approval of Lapelga®, we conducted this review of Lapelga® in the real-world setting. In addition, the biosimilar approval through Health Canada does not require efficacy data or a direct head-to-head comparison trial, so we have completed this retrospective analysis of our patients using Lapelga® in order to assess its efficacy in preventing FN. The lower reported side effects with Lapelga® in our study could be possibly related to the actual lower incidence [17], although missing information may also have contributed towards it.

The strengths of our study include the adjuvant setting with breast cancer and the variety of chemotherapy regimens used. Our sample size is reasonable to observe real-world experience. The timing of this data is appropriate as Lapelga® was approved in 2019. The weaknesses of the study include the retrospective design of the study and the data collection as there were missing pieces of information, particularly regarding the number of reported side effects related to the use of Lapelga®.

## 5. Conclusions

Lapelga® is now available as a reimbursed biosimilar, and based on our real-world experience, it showed a very similar ability to prevent FN as the original pegfilgrastim (Neulasta®).

**Author Contributions:** Conceptualization, J.Y.; methodology, J.Y.; formal analysis, J.Y.; data curation, F.K.; writing—original draft preparation, F.K., M.B. and A.C.; writing—review and editing, M.B. and J.Y.; supervision, J.Y.; project administration, M.B. All authors have read and agreed to the published version of the manuscript.

**Funding:** Unrestricted research grant funding from Apobiologix was obtained for creation of the Canadian Breast Cancer Registry Database that was used to extract the data for this retrospective study. Funders had no role in the study design, data collection and analysis, interpretation and writing, or in the decision of manuscript submission. All co-authors operate independently of the funding agencies.

**Institutional Review Board Statement:** The study was conducted in accordance with the Declaration of Helsinki, and approved by the Western University Health Science Research Ethics Board (REB#: 116627; Approved 15 October 2020).

**Informed Consent Statement:** Patient consent was waived due to the retrospective nature of this study. Data accessed was previously collected in the electronic health record and posed no additional risks to participants thus consent was approved to be waived by the research ethics board.

**Data Availability Statement:** The data presented in this study are available on request from the corresponding author.

**Acknowledgments:** Many thanks to the South West Regional Cancer Program Decision Support for gross data collection.

**Conflicts of Interest:** Unrestricted research grant funding from Apobiologix was obtained for creation of the Canadian Breast Cancer Registry Database that was used to extract the data for this retrospective study. The funders had no role in the design of the study; in the collection, analyses, or interpretation of data; in the writing of the manuscript; or in the decision to publish the results.

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
