# Peer review of "Primary Prophylaxis Lapelga® in Early Breast Cancer: A Real-World Experience"

_curroncol, doi:10.3390/curroncol30030244_

Round 1
Reviewer 1 Report (Previous Reviewer 2)
The authors have addressed the comments and the manuscript could be considered for publication.
Reviewer 2 Report (Previous Reviewer 3)
Dear Prof. Younus,
I read the article entitled: Primary prophylaxis Lapelga® in early breast cancer: a re-al-world experience. I appreciate the content and this is useful study for the concept of biosimilar drug research. However following points to be considered.
1. Please just see the references format carefully, if this is matching the journal requirements.
2. The supplementary material section should include the titles of data which can be presented upon request.
Best regards,
This manuscript is a resubmission of an earlier submission. The following is a list of the peer review reports and author responses from that submission.
Round 1
Reviewer 1 Report
The manuscript is merely a compilation of data obtained with the use of Lapelga. It would be interesting to see some correlations also in the manuscript. Eg. correlation of tumor stages with the Grade of toxicity observed.
Overall, the manuscript requires considerable modifications in terms of content before it can be considered for publication.
Author Response
Reviewer #1 Comments:
The manuscript is merely a compilation of data obtained with the use of Lapelga. It would be interesting to see some correlations also in the manuscript. Eg. correlation of tumor stages with the Grade of toxicity observed.
Overall, the manuscript requires considerable modifications in terms of content before it can be considered for publication.
Our response: Thank you very much for the comments and recommendation(s). The primary aim for this communication was to provide real-world experience of Lapelga in early breast cancer in regards to its efficacy and side effects when used as primary prophylaxis in patients treated with adjuvant chemotherapy. Future analyses could be performed, however, the low rate of febrile neutropenia in this cohort would make correlations of statistical significance challenging to calculate.
Reviewer 2 Report
The objective of the authors was to review the efficacy and tolerability of Lapelga in early breast cancer patients treated with adjuvant chemotherapy. It is a single center (London Regional Cancer Program in London, Ontario, Canada) based retrospective study and the patients data analyzed in this study had Lapelga as the primary prophylaxis. The authors have reported that early breast cancer treated with adjuvant chemotherapy and supported with Lapelga as the primary prophylaxis had low incidence (in 3/151 patients or 1.98%) of febrile neutropenia. The authors have also reported that few side effects such as fever (without neutropenia), headache, and muscle aches were reported related to the use of Lapelga and all patients were able to continue with Lapelga without switching to other options. It is interesting data but the manuscript could not be considered for publication in the current form for following reasons.
Major concerns:
1) The sample size is small. Is there any possibility for authors to include data from 2020-2021 as well instead of 2019-2020 only or data from other centers to increase the sample size. For certain groups included in the study, the sample size is too small.
2) For some reason the supplementary data files are not accessible. The url mentioned on Ln # 160 of page # 4 is incorrect. Could authors please fix the issue.
Author Response
Reviewer #2 Comments:
The objective of the authors was to review the efficacy and tolerability of Lapelga in early breast cancer patients treated with adjuvant chemotherapy. It is a single center (London Regional Cancer Program in London, Ontario, Canada) based retrospective study and the patients data analyzed in this study had Lapelga as the primary prophylaxis. The authors have reported that early breast cancer treated with adjuvant chemotherapy and supported with Lapelga as the primary prophylaxis had low incidence (in 3/151 patients or 1.98%) of febrile neutropenia. The authors have also reported that few side effects such as fever (without neutropenia), headache, and muscle aches were reported related to the use of Lapelga and all patients were able to continue with Lapelga without switching to other options. It is interesting data but the manuscript could not be considered for publication in the current form for following reasons.
Major concerns:
1) The sample size is small. Is there any possibility for authors to include data from 2020-2021 as well instead of 2019-2020 only or data from other centers to increase the sample size. For certain groups included in the study, the sample size is too small.
2) For some reason the supplementary data files are not accessible. The url mentioned on Ln # 160 of page # 4 is incorrect. Could authors please fix the issue.
Our response: Thank you very much for the review and list of major concerns. Please see below for our responses to the two concerns listed:
- Based on this concern, we amended our study end date from May 2019 to June 2022 which resulted in an additional 50 patients (33% increase in sample size) meeting eligibility criteria and being included in the communication.
- Our sincere apologies for the confusion, this issue has been resolved. The URL for supplementary data has been removed. We have updated this section to indicate that Supplementary Materials are Available upon request.
Reviewer 3 Report
Dear Dr. Younus,
On behalf of MDPI journal I reviewed the article and i found following major points to be addressed:
1. the introduction is extremely small and it need to expand with the more detailed references of the studied previously performed.
2. the tables lacks the keyword explanation e.g. the size of the tumor is indicated with Tx, T1 to T4, it must be explained in the caption of the table.
3. The criteira of the selection of data seemed very generalized, there is not stats involved in this study, which must be included.
4. the references are not written in the standard format of the journal, they must be organized.
5. the in supplementary data information, it is written that the video is included, in fact there is not supplementary file attached.
these are some very important points to be corrected before re submission.
best regrads,
Saima
Author Response
Reviewer #3 Comments:
- the introduction is extremely small and it need to expand with the more detailed references of the studied previously performed.
- the tables lacks the keyword explanation e.g. the size of the tumor is indicated with Tx, T1 to T4, it must be explained in the caption of the table.
- The criteria of the selection of data seemed very generalized, there is not stats involved in this study, which must be included.
- the references are not written in the standard format of the journal, they must be organized.
- the in supplementary data information, it is written that the video is included, in fact there is not supplementary file attached.
Our response: Thank you very much for the review and comments. Please see below for our responses to the concerns listed:
- We have expanded the introduction to include more details and references.
- Thank you very much for this recommendation. Keyword explanations and applicable definitions have been added to the table legends.
- The primary aim for this communication was to provide real-world experience of Lapelga in early breast cancer in regards to its efficacy and side effects when used as primary prophylaxis in patients treated with adjuvant chemotherapy. Future analyses could be performed, however, the low rate of febrile neutropenia in this cohort would make correlations of statistical significance challenging to calculate.
- Our sincere apologies, the references have been edited based on other publications from Current Oncology. We are more than happy to further amend these if required.
- Our apologies for the confusion, this issue has been resolved. The URL for supplementary data has been removed. We have updated this section to indicate that Supplementary Materials are Available upon request.